# Skeletal Muscle Pathogenesis in Polyglutamine Diseases

**DOI:** 10.3390/cells11132105

**Published:** 2022-07-03

**Authors:** Caterina Marchioretti, Emanuela Zuccaro, Udai Bhan Pandey, Jessica Rosati, Manuela Basso, Maria Pennuto

**Affiliations:** 1Department of Biomedical Sciences (DBS), University of Padova, 35131 Padova, Italy; caterina.marchioretti@unipd.it (C.M.); emanuela.zuccaro@unipd.it (E.Z.); 2Veneto Institute of Molecular Medicine (VIMM), 35129 Padova, Italy; 3Department of Pediatrics, Children’s Hospital of Pittsburgh, University of Pittsburgh School of Medicine, Pittsburgh, PA 15100, USA; udai@pitt.edu; 4Cellular Reprogramming Unit, Fondazione IRCCS Casa Sollievo della Sofferenza, San Giovanni Rotondo, 71100 Foggia, Italy; j.rosati@css-mendel.it; 5Department of Cellular, Computational and Integrative Biology (CIBIO), University of Trento, 38100 Trento, Italy; manuela.basso@unitn.it

**Keywords:** Huntington’s disease, spinal and bulbar muscular atrophy, spinocerebellar ataxia, skeletal muscle atrophy, polyglutamine diseases

## Abstract

Polyglutamine diseases are characterized by selective dysfunction and degeneration of specific types of neurons in the central nervous system. In addition, nonneuronal cells can also be affected as a consequence of primary degeneration or due to neuronal dysfunction. Skeletal muscle is a primary site of toxicity of polyglutamine-expanded androgen receptor, but it is also affected in other polyglutamine diseases, more likely due to neuronal dysfunction and death. Nonetheless, pathological processes occurring in skeletal muscle atrophy impact the entire body metabolism, thus actively contributing to the inexorable progression towards the late and final stages of disease. Skeletal muscle atrophy is well recapitulated in animal models of polyglutamine disease. In this review, we discuss the impact and relevance of skeletal muscle in patients affected by polyglutamine diseases and we review evidence obtained in animal models and patient-derived cells modeling skeletal muscle.

## 1. Introduction

Polyglutamine diseases are a family of nine neurodegenerative diseases that includes spinal and bulbar muscular atrophy (SBMA); Huntington’s disease (HD); dentatorubral pallidoluysian atrophy (DRPLA); and spinocerebellar ataxia (SCA) type 1, 2, 3, 6, 7, and 17 [1,2] (Figure 1, Table 1). Polyglutamine diseases are caused by expansions of the cytosine-adenine-guanine (CAG) trinucleotide repeat in the exons of specific genes. These genes code for unrelated proteins, that is, androgen receptor (AR), huntingtin (HTT), atrophin-1, ataxin-1, ataxin-2, ataxin-3, α1a-subunit of the P/Q voltage-dependent calcium channel (*CACNA1A*), ataxin-7, and the TATA-box binding protein (TBP). CAG expansions result in the production of proteins with aberrantly expanded polyglutamine tracts. All polyglutamine diseases are autosomal dominant, except SBMA, which is X-linked. Polyglutamine diseases belong to the family of brain misfolding diseases, which are a large group of neurodegenerative disorders that also includes Alzheimer’s disease, Parkinson’s disease, amyotrophic lateral sclerosis (ALS), and many others [3]. Brain misfolding diseases represent a major health burden for the entire world with an estimated number of more than 30 million patients in the next 50 years [4,5]. Brain misfolding diseases share several commonalities, such as being late-onset and progressive diseases. Symptoms typically manifest around the third to fifth decade of life, except for the juvenile forms observed in patients with very long repeats [6,7]. Another key feature is neuronal loss, and neurons seem to be extremely vulnerable in these conditions despite the fact that, very often, the disease-related proteins have ubiquitous expression and housekeeping functions in the cells [8,9]. One such example is TBP, which is the universal basal transcription factor expressed in all cell types and controlling the expression of nearly all genes. Moreover, neurodegenerative diseases are characterized by the accumulation of the disease-related proteins into inclusion bodies and aggregates inside and outside the degenerating neurons [3]. Aggregation starts in a well-defined area of the brain or spinal cord and then spreads all around the central nervous system as a function of disease progression [10]. The impact of these species on toxicity is not entirely clear. Micro-aggregates and fibrils may represent the toxic species that cause neuronal loss, and inclusion body formation may be a strategy of the cell to survive protein misfolding [11]. Insoluble species sequester cellular constituents, thus resulting in pathological conditions. Again, why neurons are so exquisitely vulnerable to protein misfolding and aggregation is not known. 

In the last few decades, it has become more and more evident that patients suffering from neurodegenerative diseases have a wide range of peripheral symptoms, including metabolic syndrome, diabetes in specific conditions, altered energy expenditure, skeletal muscle wasting, and cardiac problems. Skeletal muscle can degenerate as a primary or secondary event, in a cell-autonomous and noncell-autonomous fashion, and in response to environmental insults on subjects at risk. Skeletal muscle is so important for body homeostasis that any disruptive event occurring in this tissue does not remain silent or without consequence. This is especially important in patients suffering from neurodegenerative diseases that undergo progressive deterioration. Skeletal muscle is also important for two other reasons: first, several biomarkers are found in muscle [21,22], an aspect that is particularly relevant in disease conditions that can be misdiagnosed as others [22]. Second, intervention in the muscle is predicted to have beneficial effects not only in this tissue but also in the innervated motor neurons and the entire body homeostasis, as observed in animal models [23].

## 2. Polyglutamine Diseases: Clinical Presentation and Disease Pathogenesis

Polyglutamine proteins are unrelated to each other and carry out different functions. Most of the genes have widespread expression and some of them encode proteins with housekeeping functions, such as TBP. This makes it difficult to understand why neurons are so especially vulnerable to expanded polyglutamine, and even more intriguingly why selective populations of neurons primarily degenerate in each disease condition. Indeed, each disease condition emerges from polyglutamine expansions in different proteins suggesting that the location or context of polyglutamine repeats govern the clinical spectrum of disease manifestations. Development of animal models with targeted deletion of the genes coding for the polyglutamine disease-related proteins showed, in some cases, no phenotypes and, in most other cases, phenotypes different from those caused by expanded polyglutamine (Table 1). This evidence suggests that polyglutamine expansions cause neurodegeneration mainly through toxic gain-of-function mechanisms. It is worth noting that patients may also show symptoms that partially overlap with loss-of-function mutations of the polyglutamine protein-encoding genes. This scenario is particularly evident in SBMA, thus indicating that loss-of-function mechanisms also contribute at least in part to disease pathogenesis.

HD is characterized by the dysfunction and loss of cortico-striatal neurons (reviewed by [12]), with clinical manifestations spanning from cognitive to psychiatric and motor (chorea) symptoms. Disease progression is associated with body weight loss and progressive skeletal muscle weakness, wasting, and atrophy. HTT is widely expressed in the body, including skeletal muscle. At the subcellular level, HTT localizes to the nucleus, cytosol, endoplasmic reticulum, Golgi apparatus, and mitochondria. In the neurons, HTT has also been detected in the synaptic compartment. HTT has been implicated in several processes, from the synthesis of brain-derived neurotrophic factor (BDNF) to vesicular trafficking, ciliogenesis, and others. 

SBMA is the only polyglutamine disease that is X-linked and male-specific (reviewed by [13,24]). The reason for the sex bias of SBMA is that males have higher circulating androgen levels than females, and polyglutamine-expanded AR requires androgen binding to exert most of its neurotoxic effects. SBMA is characterized by the loss of lower motor neurons and primary involvement of skeletal muscle. Indeed, AR is highly expressed in skeletal muscle, where it works as an anabolic transcription factor that enhances muscle force and mass [25]. AR localizes to the cytosol in the inactive form, and it translocates to the nucleus upon androgen binding [13]. Interestingly, AR [26] and polyglutamine-expanded AR [27] also localize to skeletal muscle mitochondria with unknown functions. 

DRPLA is characterized by the degeneration of neurons in the dentate nucleus of the cerebellum and pallidum (reviewed by [14]). Patients present with cerebellar ataxia, chorea, epilepsy, and dementia. Atrophin-1 localizes mainly to the cytosol, but it is also present in the nucleus and is involved in protein trafficking and degradation. 

Polyglutamine ataxias are characterized by the primary loss of Purkinje cells (reviewed by [28]). Particularly interesting is the case of SCA17, as TBP regulates the expression of genes transcribed by the three eukaryotic RNA polymerases in the nucleus. 

Within the family of polyglutamine disease-causing proteins, AR and TBP are the most well-known proteins in terms of structure and function; they are both transcription factors. TBP is composed of two domains, the amino-terminal domain that is intrinsically disordered, and the carboxy-terminal domain that is highly conserved and ordered into beta-sheets motifs. TBP binds to TATA-box or is recruited by basal transcription factors to the TATA-less promoters and plays a fundamental role in the process of initiation of transcription. AR is composed of three domains. Similar to TBP, AR has an intrinsically disordered amino-terminal domain, a DNA-binding domain (two zinc fingers,) and the ligand-binding domain that forms 12 alpha-helices and two beta-sheets. How polyglutamine expansions affect TBP and AR function is not known. 

## 3. Muscle Pathology in Polyglutamine Disease Patients

Polyglutamine-expanded proteins cause damage to several tissues through cell-autonomous and noncell-autonomous mechanisms (reviewed by [29,30]). Peripheral symptoms are important components of HD manifestations, highlighting the relevance of peripheral toxicity of mutant HTT in the onset and progression of disease [31]. It is interesting to note that CAG expansions in HD patients were reported to be higher in skeletal muscle compared to lymphocytes, which may suggest why skeletal muscles are more prone to or are a primary site of the disease [32]. HD patients undergo progressive severe body mass loss, which is mainly due to skeletal muscle wasting [33]. HD patients show skeletal muscle weakness and signs of peripheral motor pathology, such as defects in eye movements and swallowing, gait abnormalities, reduced lower limb muscle strength, and dysarthria [34,35,36]. Changes in body composition and muscle wasting were not detected in a cohort of HD patients with early-to-moderate symptoms [37]. However, reduction of lean and fat body mass was reported in other HD patient cohorts as an early phenomenon [38]. Of notice, this phenotype was more prominent in male patients. In addition, an HD patient developed myopathy after running a marathon, indicating that extreme exercise that affects muscle homeostasis may anticipate or exacerbate symptoms linked to a genetic mutation at the pre-exordium stage [39]. Skeletal muscle atrophy was described in a juvenile patient suffering from DRPLA [40]. Interestingly, the skeletal muscle of this patient was characterized by the presence of lipid droplets, suggesting altered lipid metabolism, which is also a feature of SBMA skeletal muscle [41]. 

SBMA represents the only neuromuscular disease within the family of polyglutamine diseases with selective degeneration of lower motor neurons resulting in skeletal muscle atrophy [42]. SBMA patients show fatigue, muscle weakness and atrophy, fasciculations, and reduced contractility [43]. SBMA muscles are characterized by neurogenic signs, such as fiber atrophy and fiber-type grouping, and myopathic changes, such as the presence of centronucleated fibers and fiber splitting and degeneration [44]. SBMA patients have very high levels of serum creatine kinase, much more than observed in patients suffering from neurogenic atrophy [45], even years before the appearance of symptoms [46]. Interestingly, in SBMA patients, the severity of peripheral symptoms directly correlates with the length of the pathogenic CAG repeat [47,48]. Although high androgen levels are responsible for disease manifestations in SBMA, an intriguing observation was that muscle strength positively correlates with testosterone levels in patients [49]. Thus, in SBMA patients, muscle atrophy is likely to result from the combination of toxic gain-of-function and loss-of-function mechanisms. Interestingly, AR has recently been shown to interact with small mothers against decapentaplegic homolog 4 (SMAD4) and stimulate the expression of a muscle hypertrophy program, a process altered by expanded polyglutamine [50].

While skeletal muscle is primarily involved in SBMA, this tissue may degenerate in the other polyglutamine diseases as a consequence of neurodegeneration or due to metabolic alterations. Patients carrying very long CAG repeats in specific genes, namely TBP and HD, develop muscle atrophy. Large CAG expansions in the gene encoding TBP are associated with juvenile forms of SCA17 involved in reduced growth, muscle weakness and atrophy, and altered ambulation and swallowing [51,52]. HD muscle atrophy can result from the degeneration of different components of the motor unit, including the myofiber and the innervating motor neuron. Alteration of cervical and lumbar motor neurons was observed in HD mice [53,54,55]. Nonetheless, abnormalities were also detected in HD patient-derived muscle cells cultured in vitro, suggesting that some of the pathological features observed in vivo are cell-autonomous [56]. Interestingly, myotubes derived from SBMA patients showed defects in the fusion process [57]. SBMA myotubes treated with androgens had decreased number of nuclei compared to control cells and reduced size, suggesting that androgens fail to exert anabolic function in SBMA cells. Myotubes derived from HD patients revealed defective myogenic differentiation in vitro [56]. These defects were associated with mitochondrial membrane depolarization, activation of apoptotic pathways, and inclusion of body formations. Although this evidence points towards a relevant role (direct or indirect) in the clinical presentation of HD, SBMA, and SCA17, a thorough analysis of skeletal muscle pathology in the family of polyglutamine diseases is missing.

## 4. Skeletal Muscle Pathology Is Recapitulated in Mouse Models of Polyglutamine Diseases

Many of the peripheral symptoms observed in patients suffering from polyglutamine diseases are well recapitulated in R6/2 mice, which express a portion of the human HD gene under human gene promoter elements (1 kb of 5 UTR sequence and exon 1 together with ~120 CAG repeats) and represent a valuable model of HD, and knock-in mice expressing HTT-150Q [58,59], SCA17 [60], and SBMA [61,62,63]. HD mice develop diabetes and metabolic syndrome, motor dysfunction, mitochondrial abnormalities, skeletal muscle weakness and wasting, and body weight loss [64,65,66]. Some of these symptoms, namely glucose intolerance and insulin insensitivity, dyslipidemia, muscle wasting, and body weight loss, have been described also in knock-in and transgenic SBMA mice [41,67], as well as in knock-in mice with either pan- or muscle-specific expression of mutant TBP modeling juvenile forms of SCA17 [60]. HD R6/2 transgenic mice with a pathologically expanded CAG tract show motor abnormalities starting early and progressively enhancing in terms of severity and manifestations. Motor dysfunction is associated with muscle denervation, neuromuscular junction (NMJ) abnormalities, defects in muscle contraction and calcium dynamics, and loss of muscle regenerative capacity [64,68]. Different from SOD1-linked ALS, HD, SBMA, and SCA17 mice do not develop paralysis. This is consistent with a lack of structural denervation. Rather, polyglutamine disease models show mild-to-severe functional denervation, with severe structural abnormalities at the NMJ occurring at the late stage of disease and not associated with motor neuron loss, as reported in several SBMA mice [61,62]. Defects in neuromuscular transmission are associated with skeletal muscle hyperexcitability in knock-in and transgenic SBMA mice [69,70]. 

In the case of SBMA, the development of multiple animal models for conditional expression of polyglutamine-expanded AR has allowed establishing that skeletal muscle is a primary target of toxicity. Overexpression of polyglutamine-expanded AR in all tissues but skeletal muscle prevented the development of symptoms in transgenic mice, indicating that expression of the disease protein is necessary for triggering disease manifestations at least in mice [71]. Conversely, overexpression of non-expanded AR solely in skeletal muscle elicited a phenotype resembling SBMA, indicating that dysregulation of AR signaling is toxic to muscle [72]. Importantly, signs of pathology develop first in muscle and later on in the spinal cord and brainstem in knock-in SBMA mice, suggesting that SBMA may initially originate in peripheral tissues and then propagate to the central nervous system [62]. It is noteworthy that in SBMA, blood biomarkers of muscle damage correlate with disease severity [73]. 

The relevance of skeletal muscle in SBMA is underlined by the fact that skeletal muscle is a “disease modifier”. A preclinical study showed that pharmacologic intervention to silence polyglutamine-expanded AR expression with antisense oligonucleotides in peripheral tissues attenuates the phenotype of knock-in SBMA mice [74]. Similarly, strategies to promote polyglutamine-expanded AR degradation and at the same time stimulate muscle hypertrophy, namely through either activation of the insulin-like growth factor 1 signaling or treatment with the beta-agonist clenbuterol, have beneficial effects on SBMA mice [23,75]. Possibly, the intervention that promotes skeletal muscle hypertrophy may attenuate disease manifestations in other polyglutamine disease models, even if skeletal muscle atrophy results as a secondary event to neurodegeneration. Moreover, the identification of muscle biomarkers that may be more easily measured than those in the central nervous system is expected to benefit undergoing and future clinical trials.

## 5. Pathological Processes in the Skeletal Muscle in Polyglutamine Diseases

There are several reported pathological processes occurring in the skeletal muscle of patients and animal models of polyglutamine diseases. These processes are well characterized in SBMA and HD and involve deposition of misfolded proteins in the forms of aggregates and inclusion bodies, mitochondrial pathology, altered metabolism and fiber-type changes, activation of catabolic pathways, and dysregulation of gene expression, ultimately leading to muscle degeneration. 

*Aggregation and inclusion body formation* A hallmark of neurodegenerative diseases is the deposition of mutant proteins into inclusion bodies, which sequester cellular proteins, such as components of the ubiquitin-proteasome system and autophagy degradation machineries causing alterations of the cellular proteome. Inclusion bodies are present in the central nervous system of HD patients and animal models of disease. Importantly, inclusion bodies have been detected not only in neurons but also in skeletal muscle fibers and other peripheral tissues of R6/2 mice and knock-in mice expressing HTT-150Q [58,76,77], as well as in primary myotubes derived from skeletal muscle biopsies of HD patients [56]. Similarly, inclusions have been detected in the skeletal muscle of mouse models of SBMA [61,63,78,79]. Inclusions positive for mutant HTT and AR are found in the cytosol and nucleus as a single structure in most cases, and their appearance correlates with the onset of muscle atrophy (Figure 1). The role of micro-aggregates and inclusion bodies is debated, as it seems that these species may be protective, and rather diffused species may be detrimental in neurons and perhaps in nonneuronal cells [11,80]. Nonetheless, their appearance and accumulation correlate with disease severity, thus representing a component of skeletal muscle pathology and possibly a biomarker of disease. Notably, in transgenic SBMA mice, polyglutamine-expanded AR accumulates in the nucleus at 15 days of age, and it starts to form inclusions by 4 weeks of age. By 8 weeks of age, which corresponds to the onset of motor dysfunction, mutant AR forms one large inclusion body inside the myonuclei, with little or no soluble protein left in the nucleoplasm and cytosol [61]. Thus, aggregation and inclusion of body formation occurs not only in neurons but also in skeletal muscle. Protein aggregation and inclusion body formation in skeletal muscle are the hallmark of many neuromuscular diseases including inclusion body myopathy, oculopharyngeal muscular dystrophy, distal myopathies, and ALS [81,82,83]. Further investigation is needed to establish what is the relevance of deposition of misfolded polyglutamine proteins in peripheral tissues on disease onset and progression. 

*Mitochondrial pathology*. Body weight and skeletal muscle function and homeostasis are tightly linked to mitochondrial function. HD muscles are characterized by mitochondrial dysfunction with defects in oxidative phosphorylation and ATP production in patients and mouse models [84,85,86]. These deficits are detected in presymptomatic patients, indicating that they occur early and are possibly causative of muscle atrophy and energy unbalance. The relevance of skeletal muscle in body homeostasis is underlined also by the observation that the R6/2 mice present with elevated energy expenditure, reduced body weight, and increased adiposity, all defects not due to reduced food intake [56]. HD myofibers show mitochondrial depolarization, reduced oxygen consumption, cytochrome c release, oxidative stress, and induction of apoptosis [56]. ATP production in skeletal muscle mitochondria is reduced not only in HD patients but also in DRPLA patients, indicating impaired energy metabolism [65]. Mitochondrial pathology is a key component of SBMA muscle pathology, as reported in many cellular and animal models of SBMA [27,41,87,88,89,90]. Analysis of muscle specimens derived from SBMA patients revealed fibers with central core-like structures deprived of mitochondria, likely due to the disposal of defective mitochondria through mitophagy [27]. Mitochondria are depolarized at a stage that in SBMA transgenic and knock-in mice corresponds to the onset of motor dysfunction [41,61]. To further support the tight relationship between muscle and mitochondria, peroxisome proliferator-activated receptor gamma coactivator 1-alpha (PGC-1α), a key factor for mitochondrial biogenesis and function, is decreased in the muscles of HD transgenic mice and patients, and treatment to enhance PGC-1α function has beneficial effects [91,92]. PGC-1α is increased in SBMA muscle, resulting in increased mitochondrial biogenesis [41], likely to compensate for enhanced mitophagy [27]. It is noteworthy that AR localizes to mitochondria in motor neuron-derived cells and myotubes [27,87]. AR has been shown to control the expression of nuclear genes encoding mitochondrial proteins and indirectly, the translation of mitochondrial genes, thus implying a nongenomic function of AR on these organelles and suggesting pathogenetic pathways in polyglutamine diseases [93]. 

*Altered muscle metabolism and fiber-type switch*. Both SBMA [41,67] and HD [64,66,94,95] muscles undergo a glycolytic-to-oxidative metabolic switch associated with fast-to-slow fiber-type change. Loss of type IIb fibers and switch to type I fibers was also reported in a juvenile DRPLA patient [40]. The glycolytic-to-oxidative switch precedes signs of denervation and neuropathy in SBMA knock-in mice [41] and transgenic mice [61]. These metabolic alterations are associated with fiber-type changes, with progressive loss of type II glycolytic fibers and concomitant increase in type IIa and IIx fibers as the disease approaches advanced stages. Consistent with these metabolic alterations, SBMA muscle is characterized by an early down-regulation of expression of glycolytic genes and a concomitant up-regulation of oxidative genes, again long before the appearance of neurogenic signs of muscle pathology. Loss of glycolytic fibers is not a specific feature of SBMA and HD, as a similar pattern of muscle pathology is also observed in other motor neuron diseases, such as ALS [96], as well as myopathic conditions, such as Duchenne muscular dystrophy [97]. Perhaps, these metabolic changes in muscle help compensate for the loss or dysfunction of fast-fatigable motor neurons. However, in a chronic setting, they may become detrimental and primarily or secondarily contribute to disease. 

*Activation of anabolic and catabolic pathways in the skeletal muscle of polyglutamine diseases*. Skeletal muscle homeostasis is maintained by the balance between anabolic and catabolic pathways. Expression of polyglutamine-expanded AR and polyglutamine-expanded HTT results in increased protein synthesis and activation of protein kinase B (also known as Akt)/mechanistic target of rapamycin (mTOR) pathway in skeletal muscle [41,98]. Activation of these anabolic pathways likely occurs to compensate for increased energy demand. At the same time, activation of catabolic pathways leads to protein degradation via proteasome and autophagy. SBMA muscle is indeed characterized by the upregulation of expression of the atrogenes, genes that are induced upon muscle damage and atrophy and that are involved in protein degradation [99]. At the same time, genes involved in autophagy are upregulated, likely with the involvement of transcription factor EB (TFEB) [41], which is dysregulated in polyglutamine diseases [100]. Chronic activation of these pathways is associated with activation of caspases that result in cell death via apoptosis, inflammation, and atrophy [99]. 

*Gene expression dysregulation.* Polyglutamine expansions affect gene expression. Interestingly, gene expression changes are similar in the central nervous system and skeletal muscle of transgenic and knock-in HD mice [94]. Importantly, gene expression changes have been detected in skeletal muscle not only in HD mice but also in patients [66]. Among the genes dysregulated in HD and SBMA transgenic and knock-in mice, several genes code for proteins involved in skeletal muscle contractility and reactivation of expression of denervation markers associated with the loss of motor units [41,62,67,101,102]. Importantly, it was reported that HD muscles were characterized by early loss of expression of muscle chloride CIC-1 and potassium Kv3.4 channels, resulting in muscle hyperexcitability [103,104]. Notably, SBMA muscles are characterized by the altered expression of genes coding for components of the ubiquitin-proteasome system, likely contributing to the dysfunction of this pathway of protein degradation during aging [105]. A key transcription factor in muscle is myocyte enhancer factor 2 (MEF2), which is required to ensure normal muscle function and homeostasis. MEF2 is sequestered into polyglutamine-expanded AR-positive inclusion bodies, resulting in its loss of function and muscle atrophy [78]. Gene expression is dysregulated in the skeletal muscle of mouse models of SCA17 through a mechanism that involves reduced association of polyglutamine-expanded TBP with MyoD and diminished recruitment of the myogenic transcription factors at the active promoters [60]. As TBP is the universal basal transcription factor, polyglutamine expansions in this protein are likely to impact global gene expression in all tissues. 

## 6. Modeling Skeletal Muscle in a Dish

With the exception of SBMA, analysis of skeletal muscle pathology in patients suffering from polyglutamine diseases is limited by the fact that muscle biopsies are not available. Due to the severity of these conditions, muscle biopsies are not routinely performed unless recommended by the physician. Muscle biopsies give the possibility to grow primary satellite cells that can be differentiated into myoblasts and myotubes for further analysis [57,106]. An alternative approach is represented by the use of induced pluripotent stem cells (iPSCs) derived from skin biopsies of patients and healthy controls. These cells can be differentiated into satellite cells [107,108], in two-dimensional (2D) cell cultures [109], and in three-dimensional (3D) human microphysiological systems that mimic the key structural and functional properties of skeletal muscle [110,111,112]. The iPSCs obtained from patients can be differentiated into several cell types, such as motor neurons, astrocytes, microglia, medium spiny neurons, neural precursors, and retinal photoreceptors [113,114,115,116,117,118,119]. SBMA, HD, and SCA iPSC-derived neural cells exhibit defects that recapitulate the principal pathological features observed in mouse models, suggesting that these cells are an extremely relevant model to study biochemical/molecular pathways and to screen drugs with beneficial effects. Polyglutamine iPSC-derived muscle cells will give researchers much information regarding the onset of disease, progression, and cell-autonomous mechanisms. Human iPSCs represent attractive cell sources for engineering biomimetic skeletal muscle thanks to their unlimited proliferative potential, the ability to differentiate into myogenic cells, and the maintenance of pathological phenotypes. In the last decade, several culture methods for iPSC-derived myogenic differentiation have been developed, and the differentiation protocols include (1) transgene methods employing the direct reprogramming of fibroblasts or iPSCs through the overexpression of master regulators of myogenic differentiation, such as MYOD1, PAX3/7 [120,121,122,123], and (2) transgene-free methods employing the use of small molecules and growth/differentiation factors, such as inhibitors of glycogen synthase kinase-3β [124], forskolin, basic fibroblast growth factor [125], and bone morphogenic protein inhibitors [126]. These two approaches allow obtaining in vitro myogenic progenitors that can be subsequently differentiated into myotubes, ready for the study of pathological mechanisms. For disease modeling and drug development, researchers have also generated 2D myotube cultures and 3D artificial muscles [126,127,128], and these meso/macroscopic 3D constructs recapitulate the skeletal muscle architecture complexity and the representation of the resident cell populations [110,129]. Technology is making enormous progress in the differentiation of entire muscles, also thanks to the use of both bioprint techniques [130,131] and different cell types (vascular, mesodermic, adipocytic) involved in the physiological behavior of the muscle. This technology can be used to understand how polyglutamine proteins interfere with proper muscle function.

## 7. Concluding Remarks

The relevance of skeletal muscle homeostasis in neurodegenerative diseases is underlined by the fact that this tissue is a valuable therapeutic target. Intervention to ameliorate skeletal muscle metabolism and reduce atrophy is effective in animal models of polyglutamine diseases. Pharmacologic inhibition of myostatin ameliorates muscle atrophy and body weight loss, indicating that it has beneficial effects on the phenotype of HD mice [132]. Ghrelin administration to HD mice ameliorates muscle atrophy and pathology, further supporting the idea that skeletal muscle is important in HD pathogenesis and is a key target tissue for therapy development [133]. Pharmacologic strategies to reduce polyglutamine-expanded AR expression in skeletal muscle ameliorates disease in knock-in SBMA mice [74]. Genetic and pharmacologic strategies to stimulate insulin-like growth factor 1 signaling ameliorate the phenotype of transgenic SBMA mice [23,134]. Moreover, treatment of SBMA mice with the beta-agonist, clenbuterol, attenuates symptoms [75], an approach that has benefits also in SBMA patients [135,136]. Therefore, strategies to delay or attenuate skeletal muscle westing and atrophy and improve metabolism may have beneficial effects on body metabolism and central nervous system function. 

## Figures and Tables

**Figure 1 cells-11-02105-f001:**
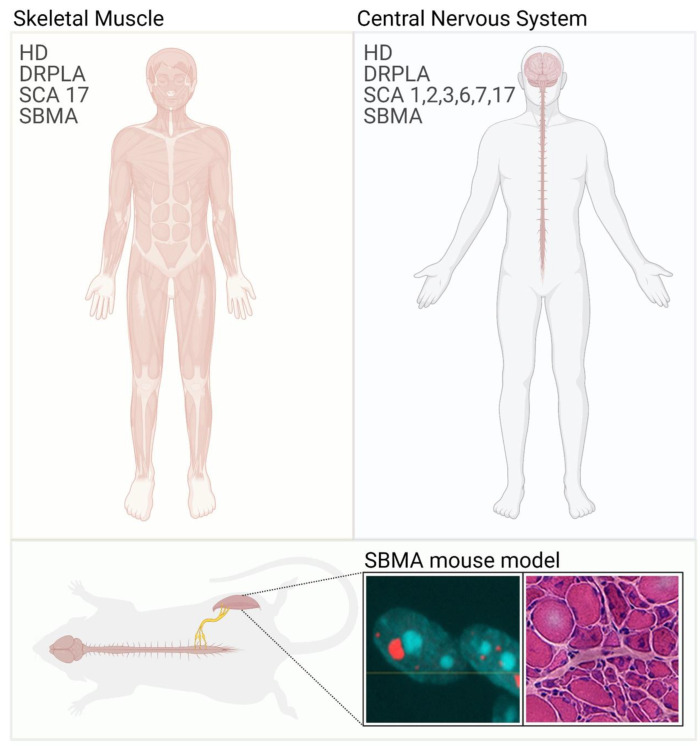
Skeletal muscle involvement in polyglutamine diseases. Polyglutamine diseases are characterized by selective neuronal dysfunction and loss (**right panel**). In some cases, patients also present with skeletal muscle atrophy and weakness (**left panel**). These symptoms are well recapitulated in animal models of polyglutamine diseases. Immunofluorescence image shows the inclusion of bodies positive for AR (red) and nuclei (blue) in myofibers isolated from SBMA mice. Hematoxylin/eosin staining of transversal sections of the quadriceps muscle of SBMA mice shows myopathy (large fibers with central nuclei) together with signs of denervation (small, angulated, and grouped fibers).

**Table 1 cells-11-02105-t001:** Polyglutamine diseases at a glance.

Disease	Gene	Normal (CAG) n	Pathogenic (CAG) n	Expression	Subcellular Localization	Function	Knock Out Mice	Vulnerable Neurons
HD [12]	*HTT*	6–35	36–39 incomplete penetrance 40–250 >75 juvenile forms	Ubiquitous	C > N	Axonal vesicular trafficking, ciliogenesis, regulation of autophagy, regulation of transcription	Embryonic lethal	Medium-sized spiny neurons in the striatum, cortical projection neurons
SBMA [13]	*AR*	5–35	36–37 low penetrance 38–72	Central nervous system, skeletal muscle, liver, adipose tissue, testis & prostate, and other tissues	Cytosolic Nuclear translocation induced by androgen binding	Steroid hormone receptor: Androgen-activated transcription factor	Viable Feminization	Brainstem and spinal cord motor neurons
DRPLA [14]	*ATN1*	7–34	49–88	Ubiquitous	C > N	Involved in protein trafficking and degradation	Viable Normal phenotype	Brainstem, cerebellum, deep midbrain
SCA1 [15]	*ATX1*	6–44	>39	Central nervous system, skeletal muscle, liver, kidney and other tissues	N in neurons, C in nonneuronal cells	Transcriptional regulation and RNA metabolism	Viable No Purkinjie cell degeneration (altered hippocampal synaptic plasticity)	Purkinje cells in the cerebellum, upper motor neurons
SCA2 [16]	*ATX2*	13–33	32–77	Brain, heart, skeletal muscle, liver, pancreas, placenta	C, ER/Golgi	RNA processing and metabolism	Viable Adult-onset obesity	Purkinje cells, brainstem and spinal cord, substantia nigra
SCA3 [17]	*ATX3*	12–40	54–89	Ubiquitous	C	Isopeptidase and deubiquitinating activity, proteasomal degradation, regulation of misfolded proteins	Viable Increased protein ubiquitination	Purkinje cells
SCA6 [18]	*CACNA1*	4–18	21–33	Neurons	PM	Subunit of voltage-gated P/Q calcium channel	Viable Ataxia, seizures, dystonia	Purkinje cells
SCA7 [19]	*ATX7*	7–19	20–35 incomplete penetrance 36 to >400	Brain, retina	N	Member of the transcriptional coactivator STAGA complex	Viable	Purkinje cells, photoreceptor cells of the retina
SCA17 [20]	*TBP*	25–44	47–66	Ubiquitous	N	Universal basal transcription factor	Embryonic lethal	Purkinje cells

The major relevant features of polyglutamine disease molecular genetics and pathophysiology. N, nucleus; C, cytosol; ER, endoplasmic reticulum; PM, plasma membrane.

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
