# Peer review of "Skeletal Muscle Pathogenesis in Polyglutamine Diseases"

_cells, 2022, doi:10.3390/cells11132105_

Round 1
Reviewer 1 Report
The article by Marchioretti et al. provides a review of muscle involvement in polyQ diseases. The review covers the topic adequately, but would be improved by minor revision as detailed below.
41: "third-to-fifth": should be spelled without hyphens
43: Sentence now sounds like long repeat arise in the subsequent generations. Please clarify the wording.
124/126: Unnecessary repetition
137-139: Unnecessary repeat of the information mentioned earlier in the same paragraph
Rows 100-106 (HTT functions) and 128-138: (AR and TBP): references needed
134: Some functional consequences of TBP polyQ expansions are known (e.g. Hsu et al. 2014; https://doi.org/10.1093/hmg/ddu410)
169: Please provide a reference for the statement that the myopathic changes affect the innervating neuron.
178–179: It could be useful to clarify that the correlation between strength and testosterone level is positive. This is counterintuitive since testosterone binding induces the toxic changes in AR and could be commented.
198: The listed defects were observed only in HD and not SBMA myotubes. Please cite correctly, and perhaps briefly discuss the results of SBMA myotubes.
216-217: The gene name for "human exon 1" should be mentioned even though it can be deduced from the context.
267: The R6/2 mouse should be introduced with a few words explanation, as the name alone doesn't tell much for an unexperienced reader.
278: "Transgenic SBMA mice", i.e. the same word order as used earlier maybe better.
281: Should be “inclusion body”
315: Mitochondrial effects of AR are not totally unknown (e.g. Bajpai 2019, PMID 30792308)
337-340: Sentence unclear: “results in increased protein synthesis - - - which in turn increases protein synthesis”
341–344: Reference needed for atrogenes
347-349: Reference needed for statement
408: Should be “constructs”
Author Response
41: "third-to-fifth": should be spelled without hyphens
We corrected the text.
43: Sentence now sounds like long repeat arise in the subsequent generations. Please clarify the wording.
We corrected the sentence to simplify text.
124/126: Unnecessary repetition
We corrected the text accordingly.
137-139: Unnecessary repeat of the information mentioned earlier in the same paragraph
Rows 100-106 (HTT functions) and 128-138: (AR and TBP): references needed
We corrected the text.
134: Some functional consequences of TBP polyQ expansions are known (e.g. Hsu et al. 2014; https://doi.org/10.1093/hmg/ddu410)
We corrected the text.
169: Please provide a reference for the statement that the myopathic changes affect the innervating neuron.
We removed the sentence as we believe it was not supported by clear evidence. Treatment targeting skeletal muscle has effects on both muscle and motor neurons. This evidence is cited in the paragraph related to animal models.
178–179: It could be useful to clarify that the correlation between strength and testosterone level is positive. This is counterintuitive since testosterone binding induces the toxic changes in AR and could be commented.
We modified the text as indicated by the reviewer. We highlighted that although androgens trigger disease, high levels in patients positively correlated with muscle strength. Thus, the disease is not only the result of a toxic GOF, but also of the AR LOF.
198: The listed defects were observed only in HD and not SBMA myotubes. Please cite correctly, and perhaps briefly discuss the results of SBMA myotubes.
We modified the text as indicated by the reviewer. We added more details on SBMA myotubes and clarified that SBMA myotubes show defects in fusion and present with reduced number of nuclei and size, suggesting that androgens do not exert anabolic functions in these cells.
216-217: The gene name for "human exon 1" should be mentioned even though it can be deduced from the context.
We corrected the text.
267: The R6/2 mouse should be introduced with a few words explanation, as the name alone doesn't tell much for an unexperienced reader.
We added details to describe this mouse model of HD (line 207).
278: "Transgenic SBMA mice", i.e. the same word order as used earlier maybe better.
We corrected the text.
281: Should be “inclusion body”
We corrected the text.
315: Mitochondrial effects of AR are not totally unknown (e.g. Bajpai 2019, PMID 30792308)
We thank the reviewer for raising this point. We added a sentence to indicate that AR localization to mitochondria can be a potential pathogenetic pathway in polyglutamine diseases due to its effect on the expression of mitochondrial genes.
337-340: Sentence unclear: “results in increased protein synthesis - - - which in turn increases protein synthesis”
We removed a part of the sentence. We apologize for this oversight.
341–344: Reference needed for atrogenes
We added the reference in support of this statement.
347-349: Reference needed for statement
We added the reference in support of this statement.
408: Should be “constructs”
We corrected the text.
Reviewer 2 Report
In this paper of Marchioretti et al. authors review current literature on polyglytamine diseases (PDs). In general, the paper is well written and authors’ ideas are clear. I have only few minor concerns and suggestions about this review:
1. Abstract needs re-writing, it says very little about the PDs
2. Authors used many abbreviations, some abbreviations are never decoded (CAG – line 30, TBP – line 46, AR – line 60)
3. List of abbreviations used would be very helpful
4. Figure 1 – not clear what authors wanted to show, legend is not helpful.
5. In several parts of the review authors miss the needed references. For example, sentences in lines 39-54, 68-76, 84-97 clearly need references to substantiate the authors’ statements.
6. Table 1 also needs references for each line.
Author Response
- Abstract needs re-writing, it says very little about the PDs
We followed the reviewer suggestion and modified the Abstract accordingly.
- Authors used many abbreviations, some abbreviations are never decoded (CAG – line 30, TBP – line 46, AR – line 60)
We spelled out these abbreviations.
- List of abbreviations used would be very helpful
We added the list of abbreviations. Moreover, we reduced the number of abbreviations, to make the text easier to read.
- Figure 1 – not clear what authors wanted to show, legend is not helpful.
We modified the Figure legend to make the message here clearer. We would like to say that skeletal muscle is a tissue involved in some polyglutamine diseases. Moreover, animal models also show muscle pathology, as in the case of SBMA.
- In several parts of the review authors miss the needed references. For example, sentences in lines 39-54, 68-76, 84-97 clearly need references to substantiate the authors’ statements.
We added references.
- Table 1 also needs references for each line.
We added references.
Reviewer 3 Report
Marchioretti et al. provide a complete and comprehensive review on the importance of skeletal muscle in polyglutamine diseases. The authors explain the relevance of muscle in terms of pathogenesis, therapeutical targeting, and monitoring of disease progression.
Two suggestions:
1. The abstract should be more focused on the issues of the review
2. Table 1: It would interesting to detail the expression in muscle of each gene
Author Response
- The abstract should be more focused on the issues of the review
We changed the Abstract as suggested also by reviewer 2. We believe that now the Abstract is more focused on the subject of this review.
- Table 1: It would be interesting to detail the expression in muscle of each gene
We added this information to the Table.